# The Prevalence of Legal Performance-Enhancing Substance Use and Potential Cognitive and or Physical Doping in German Recreational Triathletes, Assessed via the Randomised Response Technique

**DOI:** 10.3390/sports7120241

**Published:** 2019-11-26

**Authors:** Sebastian Seifarth, Pavel Dietz, Alexander C. Disch, Martin Engelhardt, Stefan Zwingenberger

**Affiliations:** 1Department of Sports Medicine at the University Center for Orthopedics and Traumatology, University Medicine Carl Gustav Carus, Technical University Dresden, 01307 Dresden, Germany; Sebsei93@aol.de (S.S.); Alexander.Disch@uniklinikum-dresden.de (A.C.D.); 2Bundeswehr Hospital Ulm, 89081 Ulm, Germany; 3Institute of Occupational, Social and Environmental Medicine, University Medical Centre of the University of Mainz, Mainz, Germany, 55131 Mainz, Germany; pdietz@uni-mainz.de; 4Department of Orthopedics, Trauma and Hand Surgery, Klinikum Osnabrück, 49076 Osnabrück, Germany; Martin.Engelhardt@klinikum-os.de

**Keywords:** doping, painkillers, triathlon, recreational athletes, risk factors, RRT

## Abstract

This study investigated the use of performance-enhancing substances in recreational triathletes who were competing in German races at distances ranging from super-sprint to long-distance, as per the International Triathlon Union. The use of legal drugs and over-the-counter supplements over the previous year, painkillers over the previous 3 months, and the potential three-month prevalence of physical doping and or cognitive doping in this group were assessed via an anonymous questionnaire. The Randomised Response Technique (RRT) was implemented for sensitive questions regarding “prescription drugs […] for the purpose of performance enhancement […] only available at a pharmacy or on the black market”. The survey did not directly state the word “doping,” but included examples of substances that could later be classed as physical and or cognitive doping. The subjects were not required to detail what they were taking. Overall, 1953 completed questionnaires were received from 3134 registered starters at six regional events—themselves involving 17 separate races—in 2017. Of the respondents, 31.8% and 11.3% admitted to the use of dietary supplements, and of painkillers during the previous three months, respectively. Potential physical doping and cognitive doping over the preceding year were reported by 7.0% (Confidence Interval CI: 4.2–9.8) and 9.4% (CI: 6.6–12.3) of triathletes. Gender, age, experience in endurance sports, and number of weekly triathlon training hours were linked to potential physical or cognitive doping. Given the potentially relevant side effects of painkiller use and physical and or cognitive doping, we recommend that educational and preventative measures for them be implemented within amateur triathlons.

## 1. Introduction

The World Anti-Doping Agency (WADA) declares doping as fundamentally contrary to the spirit of sport [1], and this is the moral basis for them producing an annual list of banned substances. These substances can be divided in two groups according to their mode of action. Physical doping agents (e.g., sympathomimetics, anabolic steroids, or erythropoietin) have a direct effect on physical aspects of the body. In contrast, cognitive doping agents (which include stimulants such as amphetamines, methylphenidate, and antidepressants) target the central nervous system. All such substances often have multiple unexpected side effects. Even over-the-counter painkillers can provoke life-threatening risks such as hyponatraemia, uncontrolled haemorrhage, and myocardial or renal infarction [2,3]. Often these side effects are unknown or simply ignored [4]. To protect the athlete’s health, it is important to avoid the non-therapeutic use of multiple substances.

Investigations into state-sponsored doping in Russia and into positive doping cases at the World Athletics Championships have recently highlighted that doping is a major problem in elite sports [5,6,7]. Given the high extent of substance abuse amongst elite athletes, it is likely that a substantial number of recreational athletes behave in the same way [8,9,10].

As in all endurance sports, a considerable amount of doping cases are reported in triathlons [11,12,13]. However, triathletes are a diverse population, with some athletes competing in a total race distance of less than 15 km, whilst others race over more than 220 km [14]. Although the latter so-called “long-distance” group only represents approximately one-tenth of all recreational triathletes [15], it is the only triathlete group thus far within which the prevalence of doping has been examined [4,11,12,16]. When such athletes were asked whether they had used banned substances over the previous 12 months, the doping prevalence was found to exceed 10% [11,12,16]. Yet the real number of substance abusers remains an estimated value. Even doping controls underestimate the actual number of abuse by a factor of 8 [17]. This, to our knowledge, is the first study to investigate the doping behaviour of recreational triathletes racing over a wide range of distances (from super-sprint up to long-distance) as per the official International Triathlon Union (ITU).

## 2. Materials and Methods

### 2.1. Sample, Ethics, Races, Procedure

Ethical approval to perform this study was obtained from the local University Ethical Committee (document number: EK 74022017). Triathletes were surveyed at six different triathlon events in central Germany in 2017. The race distances that were involved ranged from super-sprint to long-distance triathlons (Table 1).

The paper-based questionnaire was distributed from the race registration offices on the day before and on the race day, at each event. A written explanation of the study was provided at the same locale. The athletes were informed within the aforesaid explanation that the act of submitting the questionnaire implied their informed consent to participate in the study. The athletes were requested to complete the survey prior to or directly after race registration. All the forms were in the German language. The questionnaires were handed to age-group starters for all the available race distances. For this reason, the dataset obtained is more representative of the average recreational athlete than that collected by previous studies, all of which were exclusive to long-distance athletes [11,12]. For the purposes of anonymity, all the completed forms were collected in a black box and no information about the name, birth date, the distance raced, nor the estimated finish time of the participant was requested. The term “doping” was circumvented in the questionnaire in order to reduce any problems with compliance that this might cause [11]. It was replaced by the German equivalent of “prescription drugs […] with the goal of increasing (mental or physical) performance. Substances that can only be obtained from a pharmacy or on the black market”. This indirect method of questioning was chosen because it typically yields higher prevalence rates for sensitive issues and thus a more valid picture of athletes’ behaviour [12].

### 2.2. Questionnaire

At the beginning of the questionnaire, the athletes were informed both of the purpose of the survey and that participation was anonymous and voluntary. The Randomised Response Technique (RRT) was used to estimate the 12-month prevalence of prohibited substances [18]. The complete RRT question to assess the prevalence of physical doping is shown in Table 2. This assessment period was chosen, instead of life time prevalence, in order to obtain comparable data to those of previous studies. Some information was obtained via closed questions with respect to sex (male/female), A-level (i.e., possession of a German diploma that qualifies the holder for university admission, yes/no), training in a group (yes/no), ingestion of painkillers during training/competition (prophylactic/therapeutic/rest in case of pain/no pain), 12-month prevalence for the use of legal and freely available substances for physical (yes/no), and cognitive (yes/no) enhancement. The next set of questions related to biographical data (e.g., age, height, and weight) and training behaviour such as previous years of training in endurance sports, and number of weekly training hours in swimming/cycling/running. Finally, the athletes were asked which triathlon distances (out of super-sprint, sprint, Olympic-distance, half-distance, and long-distance) they had competed over within the past 12 months. The athletes were also asked to give details of painkiller intake and the underlying rationale for said intake. The athletes’ motivation for supplementation was consequently divided into prophylactic vs. therapeutic and training vs. competition related use. In order to question the gateway hypothesis, the question about dietary supplements differentiated between the intake of physical (e.g., bronchodilator) or mental (e.g., concentration-enhancing) substances. The said hypothesis states that the usage of freely available substances may lead to the later abuse of prohibited substances for the purpose of performance enhancement [19].

### 2.3. Randomised Response Technique (RRT)

RRTs are specifically developed to obtain more valid estimates when sensitive topics are studied, through their guarantee of a maximum amount of anonymity to the respondent [11,18,20]. In the present survey, a paper-and-pencil version of the unrelated question model (UQM) was used to estimate the potential prevalence of physical and cognitive doping (${\hat{\pi }}_{s}$) [21]. The UQM as it was used for the present study has been explicitly described by previous articles [11,12,22]. An explicit example of its calculation is provided by Franke et al. [23]. Similarly to their study, we used a probability for receiving the sensitive question (*p*) of 245.25/365.25. Afterwards the participants who were randomised to the sensitive group were asked a sensitive question (Questions B, Table 2), whilst the others were asked a neutral question.

The probability for answering the neutral question with “yes” ($\pi_{n}$) was 181.25/365.25. With this model, even the interviewer is unable to know whether the interviewee has answered the sensitive question or not. Furthermore, the RRT can be used to assess separate doping-prevalence values for sub-categories (e.g., females vs. males, users vs. non-users of painkillers), if the number of participants for several groups is high enough. For the purposes of this study the term “potential doping” is used for those athletes who (after completion of the RRT) gave a positive answer to the sensitive question. This definition differs from the WADA definition of doping i.e., “doping is the occurrence of one or more of the anti-doping rule violations set forth in Article 2.1 through Article 2.10 of the Code” [24]. In order to minimise the time that was taken to complete the survey, and thereby maximise compliance, we did not ask the athletes to identify what substance(s) they were taking. Strictly speaking, therefore, our data relate to the potential prevalence, rather than the actual prevalence, of doping in German recreational triathletes. Our methodology, however, allows our results to be directly compared to those of the only previously published triathlete specific studies in this area [11,12].

### 2.4. Statistics

Descriptive data are presented as mean ± SD values for continuous scaled variables and as numbers and percentages for non-continuous scaled variables. They were obtained using SPSS software, version 22. Prevalence estimates (${\hat{\pi }}_{s}$
) for physical and cognitive doping are presented as percentages with 95% confidence intervals (CI) and standard error (SE), as obtained via MATLAB version R2015a. The continuous variables “age” and “years doing endurance sports” were dichotomized by median. The splitting enabled us to calculate separate prevalence estimates, for example for younger/older athletes. Post-hoc power analyses [25] were performed for all RRT calculations, in order to test whether the sample sizes were adequate.

## 3. Results

A total of 3134 recreational athletes, on the start lists for six different triathlon events, were surveyed. Overall, 1989 (63.5%) questionnaires were received from them (Table 1 and Table 3). Of these, 1953 forms (98.2%) were sufficiently completed and evaluated. More than half of all the questionnaires (*n* = 1046; 53.6%) were collected at the Schlosstriathlon Moritzburg (sprint, Olympic, half-distance, and long-distance) event, and about one-fifth (*n* = 419; 21.5%) were obtained at the Leipzig (sprint and Olympic-distance) triathlon. The other races that were surveyed yielded smaller amounts of data: 170 forms at Powertriathlon Gera (8.7%, involving super-sprint, sprint, and Olympic-distance races); 127 forms at Koberbachtalsperre (6.5%, from super-sprint, sprint, and Olympic-distance triathlons); 106 forms at Paradiestriathlon Jena (5.4%, from super-sprint and sprint races); and 84 forms at ICAN Nordhausen (4.3%, for Olympic-distance and half-distance triathlons). Of the study participants, 76.4% (*n* = 1491) were male. The mean athlete age was 39.6 years. Subject and event characteristics are presented in Table 3.

### 3.1. Dietary Supplements and Painkillers

Of the study respondents, 31.8% declared that they had taken dietary supplements, 6.9% reported the use of cognitive enhancers and 9.7% stated that they had used physical enhancers. Intake of substances from both of the latter groups was reported by 14.2% of the athletes who took part in the study.

Painkiller use within the previous three months was reported by 11.3% of participants. We found slight differences between training and competition in the underlying rationale that was given for such intake. The prevalence of within competition painkiller use for therapeutic reasons (3.5%) was similar to that for prophylactic (3.6%) reasons. However, during training sessions more athletes used painkillers to treat their pain (4.7%), than to avoid it (2.0%). Furthermore, we identified that the intake of painkillers was associated with the use of just potential physical doping, or the use of potential physical and cognitive doping substances. More athletes used painkillers when they had raced an Olympic-distance triathlon. Additionally, we found that the use of painkillers is less likely amongst first-time starters. The use of legal and freely available substances by the study participants is summarised in Table 4.

### 3.2. RRT Results for Physical and Cognitive Doping

The survey, which used the Unrelated Question Model (UQM), demonstrated an overall prevalence of potential physical doping in its respondents over the previous year of 7.0% (CI: 4.2–9.8). The following factors increased an athletes’ prevalence for potential physical doping: more than 10 years of experience in doing endurance sports (9.4%), an age older than 39 years (9.8%), participation in an Olympic-distance event (9.3%), training more than 8 h per week (8.0%), and not training in a group (8.6%).

In comparison, the prevalence for potential cognitive doping was higher in those athletes who usually trained in a group (11.2%). Female athletes (13.2%) and athletes who did not race during the 12 months leading up to the study (15.0%) were also more willing to enhance their cognitive performance than were male or experienced athletes. The overall prevalence of potential cognitive doping was found to be 9.4% (CI: 6.6–12.3). There was an association between the intake of painkillers during the last three months and potential physical or cognitive doping. However, no significant correlation was detected between the use of dietary supplements and the use of prohibited substances. All associated influences, as well as the overall intake rate of performance-enhancing drugs, can be found in Table 5 and Table 6 and Appendix A. The overall hours of weekly training of the athletes was calculated as the summation of the weekly number of hours that they spent swimming, cycling, and running. Detailed statistical analyses are attached as Appendix A to this paper (Appendix A).

## 4. Discussion

Our study found that a significant proportion of recreational triathletes both used painkillers and potentially implemented physical and cognitive doping. Previous studies of long-distance triathletes have shown them to use substances such as anabolic steroid hormones, erythropoietin growth hormones, and amphetamines [11,17]. Such substance abuse is not only associated with triathlon. Numerous studies of professional athletic sports athletes in the late 1990s detected a prevalence of steroid abusers of 20% [9,26,27,28]. Recreational level athletes in sports such as football, athletics, tennis, handball, or gymnastics have also been reported to use physical doping [13]. Depending on the sport in question, lifetime doping prevalence values varied between 4% and 30% of athletes.

In triathlon, just two surveys in this field have been conducted to date [11,12,16]. Both examined 12-month prevalence as opposed to lifetime prevalence. They reported 10% to 18% of long-distance triathletes to have implemented potential physical doping over the year leading up to their being surveyed [11]. In our athletes, who were racing over more of the more commonly raced triathlon distances, overall potential prevalence, over an equivalent assessment period, was 7.0% for physical doping and 9.4% for cognitive doping. Our values were obtained from a questionnaire that was almost identical to that of Dietz et al. [11]. The only difference from the latter survey was the fact that, as they are legally available in some countries, we removed caffeine (pills) from the list of examples of potential cognitive doping that was provided within it. Instead, we listed caffeine as an example of an “legal and over-the-counter drug […] with the goal of increasing mental performance”. This minor difference between the two questionnaires is unlikely to account for the differences in the results that were obtained from them. The lower prevalence of doping in recreational triathletes racing super-sprint as compared to long-distance triathlons that has been reported for long-distance triathletes (i.e., 13% cognitive; 15.1% physical doping) [11,12,16] may be related to the high proportion of athletes who were performing their first triathlon (*n* = 289) in our study.

We identified five key factors to be associated with potential physical or cognitive doping. Firstly, we found that athletes who were more than 39 years of age more often reported the use of physical enhancing drugs than did younger athletes. Dietz et al. did not detect age as a predictor for physical doping in recreational long-distance triathletes [11]. Secondly, we identified that proportionally more females than males used potential cognitive doping. This finding contradicts that of Dietz et al. They reported proportionally more male than female long-distance triathletes to utilise cognitive doping [11]. Thirdly, the athletes in our study who had over 10 years of competitive endurance sports experience more often used physical doping substances than those who had less. This finding agrees with that of Dietz et al. [11]. Fourthly, we found that athletes who spent more than 8 h training per week (with the population dichotomised by median) more often used physical doping than those who trained less than 8 h in total. This relationship between weekly training time and potential susceptibility to doping has not previously been either examined or reported. Lastly, we found that, of the athletes in our sample, the Olympic-distance triathletes presented the highest prevalence of use of prohibited substances for physical doping. This is also a novel finding. Additionally, we observed that novice athletes showed the lowest tendency to enhance their performance through (potential) physical doping, and that athletes who were racing over a longer distance exhibited a higher prevalence of potential physical doping. These were not unexpected findings. Long-distance athletes may be more predisposed to feeling that they require prohibited substances, as a consequence of both greater physiological demand and greater financial costs of competition being placed upon them as compared to short-distance athletes. In contrast, we found that athletes racing shorter distances, or even those who were racing their first triathlon, tended to use potential cognitive doping more often. This could theoretically be explained by novices possessing a higher level of race-associated excitement or anxiety than experienced athletes who are competing over longer distances.

It has been hypothesised that the likelihood of future doping may be increased when an athlete uses nutritional supplements or painkillers [29]. In order to test this so called “gateway hypothesis”, we also asked our athletes whether they had used such substances. In total, 593 of our athletes (31.8%) declared the use of legal and freely available supplements to enhance their personal performance. Athletes who used nutritional supplements had a slightly higher prevalence for potential physical doping (7.7%) than athletes who had not used nutritional supplements (6.7%). The prevalence for potential cognitive doping among both populations was also the same (9.4%). However, as we were unable to detect a statistically significant association between nutritional supplementation and potential doping, our results cannot be said to support the gateway hypothesis.

In addition to the use of nutritional supplements, painkiller use is widespread amongst recreational athletes. Most common is the intake of non-steroidal anti-inflammatory drugs (NSAIDs) or other over-the-counter analgesics [3,9]. One study reported more than half of recreational marathon runners to admit to the intake of NSAIDs [2], (presumably or partly) as a means of pain avoidance [2,3,4,16,30]. A survey of participants in the IRONMAN^®^-Brazil suggested a similar prevalence value—up to 60%—for long-distance triathletes [4]. European investigations, however, have revealed remarkably lower prevalence values, at about one-third up to one-sixth of this (9.2%–20.4%) [11,16]. The 3-month prevalence for painkillers of our study was 11.2% and comparable to the results of previous European studies. Perhaps different attitudes towards the use of painkillers in general could be responsible for this extreme geographical variation in painkiller intake by triathletes.

Two of the advantages of this study are its relatively large sample size and the fact that we used the RRT to guarantee a higher level of anonymity to its respondents than might otherwise have been the case. Although failure on the part of athletes in a hurry to understand the RRT procedure could have had an impact on some answers, these factors are likely to have had an overall positive impact on the validity of our data. Nonetheless, it is important to note that, in accordance with the previous research in this field [11,12], we did not ask those of our study subjects who responded positively to the sensitive question to enumerate exactly what they were taking. Nor, so as to minimise the administration time and maximise compliance, were our study participants provided with a “self-check list” of the substances that fulfilled the WADA definition of doping at the time of survey administration. Being amateurs, our athletes were unlikely to already be well versed in the contents of the WADA list. Clearly there is a possible “trade-off” between the fact that administration of an optional survey of this type around the (compulsory) procedure of race registration could increase its subject pool, and the fact that athletes who are there may be “time-limited”. In accordance with the work of Dietz et al. [11], we took “potential doping” to be synonymous with a positive answer to the sensitive questions of the RRT i.e., whether the athlete had taken “prescription drugs […] with the goal of increasing (mental or physical) performance […] that can only be obtained from a pharmacy or on the black market”. We only appended examples of doping substances, as opposed to the full WADA list, to these sensitive questions. It is a potential limitation of our study that the fact that our prevalence values were obtained through the use of a simpler definition of doping [1] than that of WADA means that our data cannot be directly compared to any future data that are obtained in strict accordance with the WADA definition of doping.

That notwithstanding, our key findings that recreational triathletes racing over the most common triathlon distances use “prescription drugs […] that can only be obtained from a pharmacy or on the black market with the goal of increasing (mental or physical) performance” as well as pain killers to improve their performance are important. This is despite the fact that (1) doping and medical abuse are not necessarily the same thing, (2) the use of prescribed drugs as ergogenic aids is not necessarily medical abuse, and (3) the use of prescribed drugs to enhance (sporting) performance does not necessarily constitute doping. Previous research has shown that intake of any of such substances comes with the possibility of life-threatening side effects. Our work demonstrates a need to improve awareness of the risks and to reduce the intake of such substances in recreational triathletes, through the implementation of targeted education prevention and programmes.

## Figures and Tables

**Table 1 sports-07-00241-t001:** The official International Triathlon Union (ITU) distances are presented, with the associated competition distances of swimming, cycling, and running [14] as well as the event locations, where athletes of the corresponding distances were interviewed. Race location: Gera (G), Jena (J), Koberbach (K), Leipzig (L), Moritzburg (M), Nordhausen (N)**.**

Official Distance	Swim (km)	Bike (km)	Run (km)	Race Location
Super-sprint	0.4	10	2.5	G, K, J
Sprint	0.75	20	5	G, M, K, J, L, N
Olympic	1.5	40	10	G, M, K, L, N
Half-distance	1.9	90	21	M, N
Long-distance	3.9	180	42.2	M

**Table 2 sports-07-00241-t002:** The Randomised Response Technique (RRT) procedure to assess for potential physical doping and cognitive doping.

Physical doping	Please consider a certain birthday (yours, your mother’s, etc.). Is this birthday in the first third of a month (first to tenth day)? If yes, please proceed to Question A; if no, please proceed to Question B.	
Question A	Is this birthday in the first half of the year (prior to the first of July)?	
Question B	Have you taken substances to increase your physical performance within the past 12 months that are only available at a pharmacy, at the doctor’s office, or on the black market (e.g., anabolic steroids, erythropoietin, stimulants, growth hormones)?	
	Note that only you know which of the questions you will answer	
	Yes	No
Cognitive doping	Please consider a certain birthday (yours, your mother’s, etc.). Is this birthday in the first third of a month (first to tenth day)? If yes, please proceed to Question A; if no, please proceed to Question B.	
Question A	Is this birthday in the first half of the year (prior to the first of July)?	
Question B	Have you taken substances to increase your mental performance in the past 12 months that are only available at a pharmacy, at the doctor’s office, or on the black market (e.g., stimulants, cocaine, methylphenidate, antidepressants, beta-blockers, modafinil)?	
	Note that only you know which of the questions you will answer	
	Yes	No

**Table 3 sports-07-00241-t003:** The distribution of athletes from the different locations, with biographical data, and training behaviour. Race location: Gera (G), Jena (J), Koberbach (K), Leipzig (L), Moritzburg (M), Nordhausen (N).

**Race Athletes**	*n* = 3134
**Participants (Total)**	*n* = 1989
**Response Rate**	63.5%
**Location**	
**Moritzburg**	53.6% (*n* = 1046)
**Leipzig**	21.5% (*n* = 419)
**Gera**	8.7% (*n* = 170)
**Koberbach**	6.5% (*n* = 127)
**Jena**	5.4% (*n* = 106)
**Nordhausen**	4.3% (*n* = 84)
**Gender**	76.4% male (*n* = 1477)
	23.6% female (*n* = 456)
**Age in years, (mean; SD)**	18–80 (39.6 ± 10.7)
**Height cm, (mean; SD)**	150–202 (177.9 ± 8.4)
**Weight in kg, (mean; SD)**	46–130 (78.7 ± 11.6)
**BMI kg/m^2^, (mean; SD)**	Male 14.8–41.3 (24.0 ± 2.4)
	Female 14.7–34.6 (21.9 ± 2.4)
**A-Level (German diploma, qualifies the holder for university admission)**	69.2% yes (*n* = 1351)
	30.0% no (*n* = 586)
**Years of triathlon-specific training, years (mean; SD)**	0–50 (11.9 ± 9.7)
**Hours swimming/week (mean; SD)**	0–12 (1.56 ± 1.23)
**Hours bike/week, (mean; SD)**	0–20 (4.20 ± 3.00)
**Hours running/week (mean; SD)**	0–20 (2.79 ± 1.87)
**Hours of training in total (mean; SD)**	0–39 (8.56 ± 2.14)
**distances**	
**No distance raced**	14.8% (*n* = 289)
**Super-sprint (race location G, J, K)**	4.8% (*n* = 94)
**Sprint (race location G, J, K, L, M, N)**	51.2% (*n* = 999)
**Olympic (race location G, K, L, M, N)**	43.7% (*n* = 853)
**Half-Distance (race location M, N)**	24.1% (*n* = 470)
**Long-Distance (race location M)**	8.9% (*n* = 173)

**Table 4 sports-07-00241-t004:** Twelve-month prevalence for the use of legal substances and 3-month prevalence for painkillers, divided into therapeutic and prophylactic use.

**Physical Enhancement only**	9.5% (*n* = 186)	
**Cognitive enhancement only**	6.8% (*n* = 133)	
**Both**	14.0% (*n* = 274)	
**None**	68.3% (*n* = 1334)	
**Use of painkillers during the last 3 months**	11.1% (*n* = 218)	
	Prophylactic use	Therapeutic use
**During training**	2.0% (*n* = 39)	4.7% (*n* = 92)
**During competition**	3.6% (*n* = 70)	3.5% (*n* = 69)

**Table 5 sports-07-00241-t005:** Influence of the longest distance raced on potential doping prevalence. The standard error (SE) is provided. Post-hoc power analyses (Power) were performed to verify the results.

Variable (Longest Distance Raced over Last 12 Months)			Doping Prevalence ${\hat{\pi }}_{s}$ in % (Positive Answers after RRT)	SE (${\hat{\pi }}_{s}$)	Power
None	Physical doping	*n* = 287	2.7	0.034	0.22
	Cognitive doping	*n* = 298	15.2	0.038	1
Super-sprint or sprint	Physical doping	*n* = 550	6.9	0.026	0.88
	Cognitive doping	*n* = 567	11.4	0.027	1
Olympic distance	Physical doping	*n* = 475	9.3	0.029	0.96
	Cognitive doping	*n* = 494	9.8	0.028	0.98
Half-distance or long-distance	Physical doping	*n* = 491	7.6	0.028	0.9
	Cognitive doping	*n* = 509	3.5	0.026	0.42

**Table 6 sports-07-00241-t006:** Factors associated with potential doping. The continuous scaled variables that are marked. ‘^#^’ were dichotomised by median, and post-hoc power analyses (Power) were performed.

Variable			Doping Prevalence ${\hat{\pi }}_{s}$ in % (Positive Answers after RRT)	SE (${\hat{\pi }}_{s}$)	Power
Gender					
female	Physical doping	*n* = 419	5.6	0.029	0.66
	Cognitive doping	*n* = 437	13.2	0.031	1
male	Physical doping	*n* = 1381	7.5	0.016	1
	Cognitive doping	*n* = 1428	8.3	0.016	1
A-level (German diploma, qualifies the holder for university admission)					
yes	Physical doping	*n* = 1264	7.4	0.017	1
	Cognitive doping	*n* = 1306	12.0	0.018	1
no	Physical doping	*n* = 524	4.7	0.026	0.61
	Cognitive doping	*n* = 547	2.7	0.025	0.32
Years doing endurance sports ^#^					
≤10 years	Physical doping	*n* = 963	6.3	0.019	0.96
	Cognitive doping	*n* = 1000	8.0	0.019	1
>10 years	Physical doping	*n* = 658	9.4	0.024	1
	Cognitive doping	*n* = 679	10.4	0.024	1
Training in a group					
yes	Physical doping	*n* = 907	5.4	0.02	0.89
	Cognitive doping	*n* = 936	11.2	0.021	1
no	Physical doping	*n* = 883	8.6	0.021	1
	Cognitive doping	*n* = 918	8.0	0.02	1
Age ^#^					
≤39 years	Physical doping	*n* = 933	4.6	0.019	0.8
	Cognitive doping	*n* = 956	10.0	0.02	1
>39 years	Physical doping	*n* = 844	8.9	0.022	1
	Cognitive doping	*n* = 885	8.7	0.021	1
Competition performed within the last 12 months					
yes	Physical doping	*n* = 1523	7.8	0.016	1
	Cognitive doping	*n* = 1577	8.6	0.016	1
no	Physical doping	*n* = 280	2.8	0.034	0.22
	Cognitive doping	*n* = 284	15.0	0.039	1
Use of legal/freely available substances					
yes	Physical doping	*n* = 554	7.7	0.026	0.93
	Cognitive doping	*n* = 574	9.4	0.026	0.98
no	Physical doping	*n* = 1239	6.7	0.017	1
	Cognitive doping	*n* = 1284	9.4	0.017	1
Use of analgesics during the last three months					
yes	Physical doping	*n* = 198	11.8	0.045	0.88
	Cognitive doping	*n* = 209	13.5	0.056	0.84
no	Physical doping	*n* = 1605	6.4	0.015	1
	Cognitive doping	*n* = 1659	9.0	0.015	1
Overall hours of training per week ^#^					
≤8 h	Physical doping	*n* = 995	8.0	0.02	1
	Cognitive doping	*n* = 1028	13.4	0.02	1
>8 h	Physical doping	*n* = 807	5.1	0.021	0.82
	Cognitive doping	*n* = 839	4.7	0.02	0.78

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
