# Peer review of "The Prevalence of Legal Performance-Enhancing Substance Use and Potential Cognitive and or Physical Doping in German Recreational Triathletes, Assessed via the Randomised Response Technique"

_sports, 2019, doi:10.3390/sports7120241_

Round 1

Reviewer 1 Report

The authors have responded to my comments and improved the manuscript. The conclusion still has room for improvement - what is a 'relevant number' and the final statement is speculative.

Author Response

Dear Reviewer,

we thankfully received your advices towards our conclusion. 

After discussing - without any coherent result - other ways to express our astonishment about the high number of athletes substance abuse, we decided to delete the notation “relevant number” completely. 

Furthermore, we also cut the final speculative sentence, about the blood and urine testing. 

Thank you for your remarks.

Reviewer 2 Report

Use of performance-enhancing substances amongst German recreational triathletes

 General comments

Use long-distance throughout the manuscript

Use women and men or females and males

Specific comments

Line 16: local from where?

Line 23: is a common

Line 33: delete ;

Line 38: Which World Championships

Line 43: However,

Line 44: there are longer triathlon distances than 220 km, see Scand J Med Sci Sports. 2011 Dec;21(6):e82-90. doi: 10.1111/j.1600-0838.2010.01160.x. Epub 2010 Jul 6 to insert

Line 46: add a reference

Line 51: for which distances? And which level of performance?

Line 207: years of competitive

Line 224: asked instead of questioned

Line 249: triathlete instead of triathlon athletes

Author Response

Dear Reviewer, 

we thankfully received your advices towards our resubmitted manuscript and revised the mentioned points as followed:

General comments:

Long-distance is used throughout the whole manuscript. woman was not used ; “men” is changed to “males”

specific comments:

Comment: Line 16: local from where?

Answer: “local” was used to distinguish between a bigger international or smaller regional event. To avoid any misunderstanding we changed local to regional

 Comment: Line 23: is a common

Answer: changed as advised

Comment: Line 33: delete ;

Answer: changed as advised 

Comment: Line 38: Which World Championships

Answer: We referred to the World Athletics Championships. The term was changed to World Athletics Championships

Line 43: However,

Answer: “,” was added

Line 44: there are longer triathlon distances than 220 km, see Scand J Med Sci Sports. 2011 Dec;21(6):e82-90. doi: 10.1111/j.1600-0838.2010.01160.x. Epub 2010 Jul 6 to insert

Answer: We just wanted to state, that there are different distances with different demands towards a human body. We are aware of longer possible distances that could be raced, but this fact is not relevant for the thought we wanted to express. 

Line 46: add a reference

Answer: reference followed after the sentence below. But we additionally added the reference as advised 

Line 51: for which distances? And which level of performance?

Answer: investigated distances were added, as well as the description “recreational”

Line 207: years of competitive

Answer: sentence changed as advised 

Line 224: asked instead of questioned

Answer: sentence changed as advised 

 Line 249: triathlete instead of triathlon athletes

Answer: sentence changed as advised 

Furthermore, we also cut the final speculative sentence, about the blood and urine testing.  

Thank you for your remarks.

This manuscript is a resubmission of an earlier submission. The following is a list of the peer review reports and author responses from that submission.

Round 1

Reviewer 1 Report

Use of performance-enhancing substances in triathlon recreational sports 

General comments

The research took place in a distinct part in the world; I suggest adapting the title to ‘Use of performance-enhancing procedures in recreation and elite triathletes competing in different distances in Thüringen, Germany’

The manuscript needs to be checked by a native English speaker, especially from Lines 180-225

You need to define upon the first appearance in the text the terms

-          Professional athlete

-          Recreational athlete

-          Prohibited substances (give examples)

-          Physical doping (give examples)

-          Cognitive doping (give examples)

Ironman is a registered brand, so in case you want to use Ironman you need to put the ®

In case you want to express the different triathlon distances, use Olympic distance and full distance or long-distance triathlon

Specific comments

Line 44: add a reference

Line 45: you could add the definition for physical and cognitive doping

Lines 59-60: add a reference

Line 97: ()?

Table 3: define A-Level, the race distances should start with capital letters

Line 125: Olympic distance

Line 142: race

Line 153: the races are held in Thüringen, former GDR. Look for specific references such as https://www.ncbi.nlm.nih.gov/pubmed/26913904, https://www.ncbi.nlm.nih.gov/pubmed/9216474, and many more also from former Soviet Republic

Lines 160-162: reference

Lines 204 ff: this should be moved to the Introduction to explain better what you want to investigate and show

Lines 229-232: strength? Weakness? Limitations?

Lines 233-238: you need to conclude from the results of your study

Author Response

General comments

The research took place in a distinct part in the world; I suggest adapting the title to ‘Use of performance-enhancing procedures in recreation and elite triathletes competing in different distances in Thüringen, Germany’

The manuscript needs to be checked by a native English speaker, especially from Lines 180-225

You need to define upon the first appearance in the text the terms

-          Professional athlete

-          Recreational athlete

-          Prohibited substances (give examples)

-          Physical doping (give examples)

-          Cognitive doping (give examples)

Ironman is a registered brand, so in case you want to use Ironman you need to put the ®

In case you want to express the different triathlon distances, use Olympic distance and full distance or long-distance triathlon

General Answer:

Thank you for reviewing our manuscript. I assume that by a final transmitting of the manuscript, all taken remarks were included to the text of the previous manuscript submitted. This our only explanation for the numerous striking mistakes, which were present.

We carefully revised the mentioned pointes and worked over the whole manuscript to create a better understanding. Additionally, we added the definitions of the specific terms as ‘professional’ or  ‘recreational’ athletes, or ‘physical’ and ‘cognitive’ doping.  Furthermore, we also changed the title as advised. The syntax and typography were checked completely. All revisions are marked in red text color. Attached you can find the revised version of the manuscript, to your further reviewing.

Specific comments:

1. Line 44: add a reference

Response: The complete introduction was revised and got a new structure. Some references were added as advised.

1.       Line 45: you could add the definition for physical and cognitive doping

Response: The definitions were added as advised.

2.       Lines 59-60: add a reference

Response: The reference was added as advised.

3.       Line 97: ()?

Response: The missing symbol () was added.

4.       Table 3: define A-Level, the race distances should start with capital letters

Response: Definition was added as advised. Capital letters were carefully checked through the whole manuscript.

5.       Line 125: Olympic distance

Response: Modified as advised.

6.       Line 142: race

Response: Modified as advised.

7.       Line 153: the races are held in Thüringen, former GDR. Look for specific references such as https://www.ncbi.nlm.nih.gov/pubmed/26913904, https://www.ncbi.nlm.nih.gov/pubmed/9216474, and many more also from former Soviet Republic

Response: The references mentioned were not added to the manuscript to avoid a confusion of the reader. This study concentrates on recreational athletes, while the studies above were based on professional athletes.

8.       Lines 160-162: reference

Response: The complete discussion was overworked, and references added.

9.       Lines 204 ff: this should be moved to the Introduction to explain better what you want to investigate and show

Response: The mentioned points were added to the introduction as advised.

10.   Lines 229-232: strength? Weakness? Limitations?

Response: The complete discussion was overworked, hereby also the presentation of strength and weakness of the study was revised.  

11.   Lines 233-238: you need to conclude from the results of your study

Response: The conclusion now tries to concentrate on the key findings of this study:

Concluding, this study investigated the prevalence of doping substances among triathlons for the first time, considering the different raced distances of the athletes. As expected, there is an alarming number of recreational athletes using prohibited substances. Compared to previous studies, we could show that an average triathlete has a lower prevalence for taking doping substances than an athlete racing an IRONMAN® -race. Associations between the intake of painkillers or nutritional supplements and doping were not detected. But the observed effect, that different substances were taken by athletes racing on different distances, is a completely new finding, that need further investigations to be completely understood. 

Reviewer 2 Report

General Comments: The manuscript aims to determine the prevalence of physical and cognitive doping among recreational triathletes and to determine the predictors that influence doping behaviours in this group. The manuscript has numerous irregularities in syntax and many typographical errors. The topic of this study is of interest and worth investigation; however, the document requires work to justify the aims, methods and interpretation of these findings. I provide some specific examples below, but the manuscript requires a truly thorough overall.

Specific Comments:

1. Throughout: Numerous errors exist in syntax and English language that seriously distract the reader. These examples start in the first sentence of the paragraph and continue to the last paragraph of the manuscript. For example, abstract (line 14-15) sentence 1 does not make sense and requires revising; “the aim of this study was to determine the prevalence of physical doping, cognitive doping, and use of painkillers among recreational triathletes.” I suggest that the authors should consider professional assistance to proof read and improve the English language prior to further consideration.

2. Abstract (lines 20-22 and 22-24): These sentences do not make sense – please rewrite.

3. Abstract (line 28): change “forbidden” to “banned”.

4. Introduction (line 35): remove second “also” in the sentence

5. Introduction (line 36): remove “already”

6. Introduction (line 38): spelling error “therefore”

7. Introduction (lines 40-41): reword “The real number of how often …..” This does not make sense as written.

8. Introduction: There is a reasonable body of literature in this space. The authors have not described what we currently understand about this topic and the rationale for this work. For example, if the rationale is based around the population and event types, this is not well articulated to the reader. The introduction needs to leave the reader that this work is clearly warranted!

9. Methods: The authors confuse tense in this section – please stick to the past tense in all locations (e.g., line 67).

10. Tables (throughout): Tables do not have headings and are poorly formatted throughout. Please use standard scientific presentation (decimal points, headings, abbreviations with explanations, etc).

11. Methods (lines 88-89): Insufficient detail is provided on the key outcome measures (the reader is left to consult other literature to obtain the equations that develop the key outcome measures. In my opinion, this is not appropriate – please provide this detail here.

12. Methods (line 97): “()” what does this mean?

13. Results (line 118-128): line 118: should this be “freely”? The remainder of the paragraph needs rewording as most of the sentences do not make sense.

14. Results (line 145-146): reword this sentence: “It was noticed, …, but neither …….”

15. Table 5: what are ‘ours’?

16. Discussion: the whole discussion needs an English language overhaul prior to further comment. In addition, the key findings are not systematically discussed; therefore, it is difficult to interpret what this work adds to the literature.

17. The concluding paragraph (lines 133-138) just repeats the findings of the study. Please refocus this section so that the reader can understand what this work adds. 

Author Response

General Comments:

The manuscript aims to determine the prevalence of physical and cognitive doping among recreational triathletes and to determine the predictors that influence doping behaviours in this group. The manuscript has numerous irregularities in syntax and many typographical errors. The topic of this study is of interest and worth investigation; however, the document requires work to justify the aims, methods and interpretation of these findings. I provide some specific examples below, but the manuscript requires a truly thorough overall.

General Answer:

Thank you for reviewing our manuscript. I assume that by a final transmitting of the manuscript, all taken remarks were included to the text of the previous manuscript submitted. This our only explanation for the numerous striking mistakes, which were present.

We carefully revised the mentioned pointes and worked over the whole manuscript to create a better understanding. Additionally, we added the definitions of the specific terms as ‘professional’ or  ‘recreational’ athletes, or ‘physical’ and ‘cognitive’ doping.  The syntax and typography were checked completely. All revisions are marked in red text color. Attached you can find the revised version of the manuscript, to your further reviewing.

Specific Comments:

1.       Throughout: Numerous errors exist in syntax and English language that seriously distract the reader. These examples start in the first sentence of the paragraph and continue to the last paragraph of the manuscript. For example, abstract (line 14-15) sentence 1 does not make sense and requires revising; “the aim of this study was to determine the prevalence of physical doping, cognitive doping, and use of painkillers among recreational triathletes.” I suggest that the authors should consider professional assistance to proof read and improve the English language prior to further consideration.

Response: The syntax and typography were checked thorough the whole manuscript. Almost every part of the manuscript was revised to offer a better understanding of the investigation performed. All revisions are marked in red text color.

2. Abstract (lines 20-22 and 22-24): These sentences do not make sense – please rewrite.

Response: The sentence was revised as followed:

-          11.3% of all athletes admitted the use of painkillers during the last three months.

-      Furthermore, we surveyed the reasons for this intake. Hereby, we distinguished between ‘therapy’/ ‘prophylaxis’ and ‘training’/ ‘competitional use’.

3. Abstract (line 28): change “forbidden” to “banned”.

Response: Modified as advised.

4. Introduction (line 35): remove second “also” in the sentence

Response: Modified as advised.

5. Introduction (line 36): remove “already”

Response:  Word removed as advised.

6. Introduction (line 38): spelling error “therefore”

Response: Word removed as advised.

7. Introduction (lines 40-41): reword “The real number of how often …..” This does not make sense as written.

Response: The sentence was revised as followed:

But the real number of substance abusers remains as an estimated value, as probably even doping controls underestimate the actual number of abuse by 8 times [20].

8. Introduction: There is a reasonable body of literature in this space. The authors have not described what we currently understand about this topic and the rationale for this work. For example, if the rationale is based around the population and event types, this is not well articulated to the reader. The introduction needs to leave the reader that this work is clearly warranted!

Response: The introduction was completely revised, and a new structure was given to the mentioned points to guarantee a better understanding of the aim of the study.

9. Methods: The authors confuse tense in this section – please stick to the past tense in all locations (e.g., line 67).

Response: We carefully revised the whole manuscript and took care about the used tenses.

10. Tables (throughout): Tables do not have headings and are poorly formatted throughout. Please use standard scientific presentation (decimal points, headings, abbreviations with explanations, etc).

Response: Headings, abbreviations and explanations were added as advised to the presented tables.

 11. Methods (lines 88-89): Insufficient detail is provided on the key outcome measures (the reader is left to consult other literature to obtain the equations that develop the key outcome measures. In my opinion, this is not appropriate – please provide this detail here.

Response: Further information is given as followed:

Similar to their study, we used a probability for receiving the sensitive question p of 245.25/365.25 (“Is a randomly chosen birthday within the first 20 days of the chosen month?”). Afterwards participants randomized to the sensitive group were asked a sensitive question (“Have you used doping substances in the last 12 months?”), while the others were asked a neutral question (“Is the chosen birthday in the first half of the year?”). The probability for answering the neutral question with ‘yes’ () was 181.25/365.25. By this model even the interviewer is not able to know, whether the interviewee answered the sensitive question or not. The doping prevalencecan be calculated, after subtracting the number of “necessary” yes-answers () from all given “yes” answers.
Headings, 12. Methods (line 97): “()” what does this mean?

13. Results (line 118-128): line 118: should this be “freely”? The remainder of the paragraph needs rewording as most of the sentences do not make sense.

Response: The paragraph was completely revised as advised.

14. Results (line 145-146): reword this sentence: “It was noticed, …, but neither …….”

Response: Sentence was reworded as followed:

It was noticed, that there was a relation between the intake of painkillers during the last three months and physical or cognitive doping. But no correlation between the use of dietary supplements and prohibited substances was found.

15. Table 5: what are ‘ours’?

Response: The ‘h’ was lost – it should be ‘hours’

16. Discussion: the whole discussion needs an English language overhaul prior to further comment. In addition, the key findings are not systematically discussed; therefore, it is difficult to interpret what this work adds to the literature.

Response: The whole discussion was revised and checked by a native English speaker as advised.

17. The concluding paragraph (lines 133-138) just repeats the findings of the study. Please refocus this section so that the reader can understand what this work adds.

Response: The conclusion now tries to concentrate on the key findings of this study:#

Concluding this study investigated the prevalence of doping substances among triathlons for the first time, considering the different raced distances of the athletes. As expected, there is an alarming number of recreational athletes using prohibited substances. Compared to previous studies, we could show that an average triathlete has a lower prevalence for taking doping substances than an athlete racing an IRONMAN® -race. Associations between the intake of painkillers or nutritional supplements and doping were not detected. But the observed effect, that different substances were taken by athletes racing on different distances, is a completely new finding, that need further investigations to be completely understood. 

Round 2

Reviewer 2 Report

General Comments: The revised manuscript is improved, but the quality of writing is still poor. I provide detail about some suggested edits, but these are only examples. The remainder of the manuscript still requires considerable editing to meet the required standard for publication.

Specific Comments:

1. Throughout: numerous errors exist in English language that seriously distract the reader.

2. Abstract (lines 14): I suggest that you start the first sentence with “The”.

3. Abstract (line 15): I suggest that you remove “Therefore”.

4. Abstract (line 16): I suggest that you remove “therefore” and change “to” to with.

5. Abstract (line 17): I suggest that you remove “By means of this questionnaire” and start the sentence with “Age”.

6. Abstract (line 18): I suggest that you change “interviewed” to “surveyed”.

7. Abstract (line 23): this sentence is redundant and should be removed.

8. Abstract (line 24): I am not sure what this sentence is intended to add – reword.

9. Abstract (line 26-27): this sentence should be amended (it is impossible to know whether doping enhances physical and cognitive performance.

10. Abstract (line 28-30): It is difficult to see how your data leads to this statement – I suggest that you should be cautious here about specific recommendations where you do not have data to support statements.

11. Introduction (lines 36 and 38): place a comma before “e.g.,”

12. Introduction (line 40): I suggest that you change “drugs” to “substances”.

13. Introduction (line 51): this sentence should be reworded, “vulnerable for doping” is not appropriate terminology.

14. Introduction (line 57-58): please reword this sentence.

15. Introduction (line 59): please reword this sentence to provide the reader with a clear aim statement. “this study tried to give a first idea” should be avoided.

16. Introduction (line 60-62): please remove these sentences as they relate to the method rather than aim of this work and they are poorly written.

17. Methods (line 74): reword “has been put up” with “was provided at …”

18. Methods (line 75): use past-tense please

19. Methods (line 77-78) please reword this statement

20. Methods (line 98): reword “we divided reasons for their use” needs rewording

21. Results (line 148): remove comma

22. Results (line 150): place a comma before “e.g.,”

23. Results (line 158): reword “Remarkable more …….” Does not make sense.

24. Results (line 159-160): again reword “Further we investigate, ….” Does not make sense.

25. Results (line 172): “The survey using the ….” Does not make sense – reword.

26. Results (line 182): remove comma

27. Results (line 183): insert ‘existed’ or similar after “But no correlation” and should be a very weak correlation or a non-significant correlation.

28. Discussion (line 193): “it is long been known” – reword

29. Discussion (line 196): “until today ……” does not make sense.

30. Discussion (line 197): reword.

31: Discussion (line 201): “publicized” is the wrong word – do you mean published.

32. Discussion (line 202): coma after “knowledge”

33. Discussion (line 210): change ‘of’ to ‘in’

34. Discussion (line 211): “was’ to ‘were’

35. Discussion (line 216): reword this sentence.

36. Discussion (line 223): ‘which’ should be ‘who’

37. Discussion (line 225): what trend and why is it interesting?

38. Discussion (line 232-233): this sentence adds nothing for the reader.

39. Discussion (line 237): “to proof that …” – reword

40. Discussion (line 241): what is the “non-supplying population” – you don’t have information on suppliers.

41. Discussion (line 251-): “the found prevalence” – reword

42. Discussion (lines 265-): reword

43. Discussion (line 268-): “concluding ….” reword

Author Response

General Comments: The revised manuscript is improved, but the quality of writing is still poor. I provide detail about some suggested edits, but these are only examples. The remainder of the manuscript still requires considerable editing to meet the required standard for publication.

General Response:

We are received your comments to our submitted manuscript and carefully revised the mentioned points step by step. The whole manuscript was checked by an Australian native speaker once again. We hope, that it meets the required standard for publication now.

Specific Comments:

1. Throughout: numerous errors exist in English language that seriously distract the reader.

 Response: The whole manuscript was checked by an Australian native speaker again. We hope, that it meets the required standard for publication now.

 2. Abstract (lines 14): I suggest that you start the first sentence with “The”.

 Response: Revised as advised.

 3. Abstract (line 15): I suggest that you remove “Therefore”.

 Response: Word removed as advised.

 4. Abstract (line 16): I suggest that you remove “therefore” and change “to” to with.

 Response: Revised as advised.

 5. Abstract (line 17): I suggest that you remove “By means of this questionnaire” and start the sentence with “Age”.

 Response: Revised as advised.

 6. Abstract (line 18): I suggest that you change “interviewed” to “surveyed”.

 Response: Revised as advised.

 7. Abstract (line 23): this sentence is redundant and should be removed.

 Response: Sentence removed as advised.

 8. Abstract (line 24): I am not sure what this sentence is intended to add – reword.

 Response: Sentence reworded to:

“Hereby we distinguished between different motivations for the intake. (‘therapy’/‘prophylaxis’ and ‘training’/‘competition’)”

 9. Abstract (line 26-27): this sentence should be amended (it is impossible to know whether doping enhances physical and cognitive performance.

 Response: Sentence reworded to:

“There is an alarming rate of recreational triathletes, who are trying to enhance their physical and cognitive performance by the intake of doping substances.”

 10. Abstract (line 28-30): It is difficult to see how your data leads to this statement – I suggest that you should be cautious here about specific recommendations where you do not have data to support statements.

 Response: Sentence was removed.

 11. Introduction (lines 36 and 38): place a comma before “e.g.,”

 Response: Revised as advised.

 12. Introduction (line 40): I suggest that you change “drugs” to “substances”.

 Response: Revised as advised.

 13. Introduction (line 51): this sentence should be reworded, “vulnerable for doping” is not appropriate terminology.

 Response: Sentence revised to:

“Like in all endurance sports, there is a considerable amount of doping cases reported in triathlon.”

 14. Introduction (line 57-58): please reword this sentence.

 Response: Sentence reworded to:

“The found doping prevalence was more than 10%, when athletes were asked, if they had used banned substances in the previous 12 months.”

 15. Introduction (line 59): please reword this sentence to provide the reader with a clear aim statement. “this study tried to give a first idea” should be avoided.

 Response: Sentence revised to:

“This study investigated the doping behavior of age-group triathletes in general.”

 16. Introduction (line 60-62): please remove these sentences as they relate to the method rather than aim of this work and they are poorly written.

 Response: Removed as advised.

 17. Methods (line 74): reword “has been put up” with “was provided at …”

Revised as advised.

 18. Methods (line 75): use past-tense please

 Response: Revised as advised.

 19. Methods (line 77-78) please reword this statement

 Response: Reworded to:

“Exclusively questionnaires of age-group athletes were included to the data.”

 20. Methods (line 98): reword “we divided reasons for their use” needs rewording

 Response: Reworded to:

By asking for the intake of painkillers, the reasons for their use were divided. (prophylactic vs. therapeutic and during training vs. during competition)

 21. Results (line 148): remove comma

 Response: Revised as advised.

 22. Results (line 150): place a comma before “e.g.,”

 Response: Revised as advised.

 23. Results (line 158): reword “Remarkable more …….” Does not make sense.

 Response: Revised to:

“More athletes used painkillers when they had raced an Olympic distance race.”

 24. Results (line 159-160): again reword “Further we investigate, ….” Does not make sense.

 Response: Revised to:

Additionally, we investigated, that the use of painkillers is less likely among first-time starters.

 25. Results (line 172): “The survey using the ….” Does not make sense – reword.

 Response: Revised to:

“The survey, which is using the Unrelated Question Model (UQM), resulted in an overall prevalence for athlete’s physical doping of 7.0% (CI: 4.2-9.8).”

 26. Results (line 182): remove comma

 Response: Removed as advised.

 27. Results (line 183): insert ‘existed’ or similar after “But no correlation” and should be a very weak correlation or a non-significant correlation.

 Response: Revised to:

“But no significant correlation existed between the use of dietary supplements and prohibited substances.”

28. Discussion (line 193): “it is long been known” – reword

 Response: Revised to:

Due to the fact, that doping is also used in recreational sports, this study was designed to identify the substitution behavior of dietary supplements, painkillers and doping substances among these recreational triathletes

 29. Discussion (line 196): “until today ……” does not make sense.

 Response: Revised to:

“Actually…”

 30. Discussion (line 197): reword.

 Response: Reworded to:

Numerous studies in the late 90’s had detected a prevalence of 20% of steroid abusers among fitness athletes.

 31: Discussion (line 201): “publicized” is the wrong word – do you mean published.

 Response: Revised as advised.

 32. Discussion (line 202): coma after “knowledge”

 Response: Revised as advised.

 33. Discussion (line 210): change ‘of’ to ‘in’

 Response: Revised as advised.

 34. Discussion (line 211): “was’ to ‘were’

 Response: Revised as advised.

 35. Discussion (line 216): reword this sentence.

 Response: Revised to:

“More often, athletes which were older than 39 years, reported the use of performance enhancing drugs.”

 36. Discussion (line 223): ‘which’ should be ‘who’

 Response: Revised as advised.

37. Discussion (line 225): what trend and why is it interesting?

 Response: Explained in the sentence below – athletes who raced different distances used different substances:

“First timers showed the lowest tendency to enhance their performance by physical doping, while we observed that long distance athletes showed a higher prevalence for physical doping. […] Contrary to this, we recognized that athletes racing shorter distances, or even racing their first triathlon, tended to use cognitive doping more often. […]”

 38. Discussion (line 232-233): this sentence adds nothing for the reader.

 Response: Removed as advised.

 39. Discussion (line 237): “to proof that …” – reword

 Response: Revised to:

“Therefore, we also questioned our athletes, if they had used such substances.”

 40. Discussion (line 241): what is the “non-supplying population” – you don’t have information on suppliers.

 Response: Revised to:

“Athletes who used nutritional supplements had a slightly higher prevalence for physical doping (7.7%) than athletes who hadn’t used nutritional supplements (6.7%).”

 41. Discussion (line 251-): “the found prevalence” – reword

 Response: Revised to:

“The 3-month prevalence for painkillers of this study was found to be 11.2% and […]”

 42. Discussion (lines 265-): reword

 Response: Revised to:

No valid outcome could be obtained for these subpopulations, due to a too small population.

 43. Discussion (line 268-): “concluding ….” reword

 Response: Revised to:

“This study investigated the prevalence of doping substances among triathletes, who were racing different distances, for the first time.”